# Detection and Typing of Human Enteroviruses from Clinical Samples by Entire-Capsid Next Generation Sequencing

**DOI:** 10.3390/v13040641

**Published:** 2021-04-08

**Authors:** Manasi Majumdar, Cristina Celma, Elaine Pegg, Krunal Polra, Jake Dunning, Javier Martin

**Affiliations:** 1Division of Virology, National Institute for Biological Standards and Control (NIBSC), Potters Bar, Hertfordshire EN6 3QG, UK; manasi.majumdar@nibsc.org (M.M.); elaine.pegg@nibsc.org (E.P.); 2Enteric Virus Unit, Public Health England, London NW9 5EQ, UK; Cristina.Celma@phe.gov.uk (C.C.); k.polra@imperial.ac.uk (K.P.); Jake.Dunning@phe.gov.uk (J.D.)

**Keywords:** enterovirus surveillance, human enterovirus, next generation sequencing (NGS), direct detection, clinical diagnosis, whole-genome sequencing

## Abstract

There are increasing concerns of infections by enteroviruses (EVs) causing severe disease in humans. EV diagnostic laboratory methods show differences in sensitivity and specificity as well as the level of genetic information provided. We examined a detection method for EVs based on next generation sequencing (NGS) analysis of amplicons covering the entire capsid coding region directly synthesized from clinical samples. One hundred and twelve clinical samples from England; previously shown to be positive for EVs, were analyzed. There was high concordance between the results obtained by the new NGS approach and those from the conventional Sanger method used originally with agreement in the serotypes identified in the 83 samples that were typed by both methods. The sensitivity and specificity of the NGS method compared to those of the conventional Sanger sequencing typing assay were 94.74% (95% confidence interval, 73.97% to 99.87%) and 97.85% (92.45% to 99.74%) for Enterovirus A, 93.75% (82.80% to 98.69%) and 89.06% (78.75% to 95.49%) for Enterovirus B, 100% (59.04% to 100%) and 98.10% (93.29% to 99.77%) for Enterovirus C, and 100% (75.29% to 100%) and 100% (96.34% to 100%) for Enterovirus D. The NGS method identified five EVs in previously untyped samples as well as additional viruses in some samples, indicating co-infection. This method can be easily expanded to generate whole-genome EV sequences as we show here for EV-D68. Information from capsid and whole-genome sequences is critical to help identifying the genetic basis for changes in viral properties and establishing accurate spatial-temporal associations between EV strains of public health relevance.

## 1. Introduction

Virus outbreaks are a constant threat to our health systems and monitoring circulating strains is extremely relevant for contingency planning, outbreak management, and containment response. Controlling viral diseases has become a health priority and the World Health Organization (WHO) set an ambitious target to eradicate poliomyelitis disease. The Global Polio Eradication Initiative requires the essential support of a surveillance program that can sensitively detect polio (PV) and non-polio enterovirus (NPEV) circulation [1].

EVs are members of the enterovirus genus of the family *Picornaviridae* infecting humans and are classified into four species (A-D) based on genetic divergence [2]. EVs are now recognized as the most common cause of meningitis and can cause other severe diseases including myocarditis, sepsis like syndrome, respiratory diseases, and acute hepatitis [3]. The recent evidence of increased detection of enterovirus D68 (EV-D68) in respiratory samples and the temporal and geographical association of these outbreaks with an increase in acute flaccid myelitis (AFM) cases observed in the United States and Europe [4,5], as well as periodic outbreaks of EV71 in Asia, demonstrates the significant morbidity and mortality that can be caused by EVs [6]. Effective monitoring and early detection combined with genetic characterization of EVs in appropriately collected samples are crucial [7]. Conventional Sanger sequencing of part of the gene encoding VP1 capsid protein has been the gold standard for genomic analysis and genotyping of EVs in public health laboratories for decades [8,9]. This methodology has been shown to be far more sensitive than traditional virus isolation using cell cultures. However, the expansion of next generation sequencing (NGS) technology has exponentially increased the genomic information that can be gathered from pathogens. These new NGS approaches have improved molecular epidemiology resolution, primer designing, studying genomic recombination events, pathogens identification and association with syndromes where etiologies often remain unknown like encephalitis, fulminant hepatitis, sepsis, etc., [10,11,12,13,14,15,16,17,18,19]. 

We have assessed an NGS approach that generates EV whole-capsid nucleotide sequences as a high throughput state-of-art diagnostic method for EV detection and identification in clinical samples. The advantages of using this NGS approach over conventional methods relying on short PCR Sanger sequences are discussed. We show the importance of NGS technology for the molecular dissection of EV strains, including the possibility to easily expand the technique to obtain whole-genome sequences of target EV serotypes (Appendix A).

## 2. Materials and Methods

### 2.1. Sample Selection

A total of 112 clinical samples from the period June 2017 to September 2018 that had tested positive for EV at Public Health England (PHE) using a reverse transcription (RT)-real time quantitative polymerase chain reaction (RT-qPCR) assay were selected for this study (Appendix A). The studied samples represented a variety of clinical specimens: cerebrospinal fluid (CSF, *n* = 39, 34.82%), respiratory (*n* = 36, 32.14%), stool (*n* = 25, 22.32%), blood (*n* = 6, 5.36%), 4 skin and vesicle swabs, and 2 samples with unspecific origin (Table 1). The study panel included samples containing a broad range of virus loads, inferred from the range of Ct values in the RT-qPCR assay, from 10 to 40 (25% percentile Ct 22.53; 75% percentile Ct 30.64). EV strains from all four A (*n* = 19), B (*n* = 48), C (*n* = 7), and D (*n* = 13) EV species, as typed using a conventional Sanger sequencing method, were included in the study. Among them, some samples contained newly emerging EV serotypes such as EV-D68, EV-C104, EV-C105, EV-A89 and some were untypeable by the conventional method (*n* = 25).

### 2.2. RNA Extraction, EV Detection, and Conventional Typing PCR at PHE

As part of an enhanced EV surveillance, laboratories in England are requested to submit positive EV samples to PHE for confirmation analysis and genotyping. Clinical samples included in this study were tested using a routine EV RT-qPCR assay to determine the presence of EV RNA as described before. Partial amplification of the genome 5′end was performed using an in-house assay. Briefly, nucleic acid was extracted from 200 μL of clinical material (10% suspensions in case of fecal samples) using an automatic RNA extraction platform (Qiasymphony, Qiagen, Dusseldorf, Germany) or by manual extraction (QiaAmp viral RNA, Qiagen, Dusseldorf, Germany). EV detection was performed using Fast Virus 1-Step PCR Mastermix (Invitrogen) with primers EV-F 5′-GCCCCTGAATGCGGCTAA T-3′, EV-R 5′-AAACACGGACACCCAAAG TA-3′ and probe EV-Pr 5′-6-FAM-TCT GYR GCGGAACCGACT-MGB-3′. Mengovirus was used as an internal process control (added before the nucleic acid extraction step) and detected using primers MengoF: 5′ GCGGGTCCTGCCGAAAGT-3′, MengoR 5′-GAA GTAACATATAGACAGACGCACAC-3′ and probe: MengoP5′-VIC-ATCACATTACTG GCCGAAGC-MGB-3′. Cycle conditions were 50 °C for 15 min and 95 °C for 2 min followed by 45 cycles of 95 °C for 15 s and 60 °C for 60 s. EV-positive samples were genotyped using an EV typing assay recommended by WHO [20] referred here as the conventional EV typing assay. This typing assay consists of an RT step with a set of specific primers followed by seminested PCR amplification as described elsewhere [8]. Purified DNA products of about 300 bp in length, corresponding to a partial VP1 coding sequence, were sequenced using an ABI Prism 3130 genetic analyzer (Applied Biosystems, Foster City, CA, USA). 

### 2.3. Modified Pan-EV Entire-Capsid Coding Region RT-PCR Amplification (mECRA)

We followed a method we recently described which is designed to amplify entire-capsid sequences of EV strains from all four Enterovirus A, B, C, and D species [16], which was modified from an original method described before, primarily designed to amplify PV sequences [21]. Briefly, two independent PCR reactions were performed using two different primer sets: 

1st set of primers 5′NCR (5′-TGGCGGAACCGACTACTTTGGGTG-3′) and CRE-R (5′-TCAATACGGTGTTTGCTCTTGAACTG-3′).

2nd set of primers MM_EV_F2 (5′-CAGCGGAACCGACTACTTT-3′) and MM_EV_R1 (5′-AATACGGCATTTGGACTTGAACTGT-3′).

Reaction conditions were: 50 °C for 30 min followed by 94 °C for 2 min plus 42 cycles of 94 °C for 15 s, 55 °C for 30 s, and 68 °C for 8 min with a final extension step of 68 °C for 5 min. Amplified products from both reactions were purified using AMPure XP magnetic beads (Beckman Coulter, Brea, CA, USA) and pooled (1:1) before being analyzed by NGS. The expected amplicon size for both RT-PCR is approximately 4000 nucleotides (nucleotides 553–4459, numbering as in PV1 Sabin AY184219 reference strain). Our method, in combination of NGS analysis described in Section 2.5 below, can discriminate between genotypes and sub-genotypes within serotypes and has been extensively validated using laboratory mixtures of reference EV strains of known sequence and comparison with Sanger sequence analysis and analysis using the Oxford Nanopore MinION system which allows sequencing complete PCR products [18].

### 2.4. Whole-Genome Amplification and Sequencing of EV-D68 Strains from Clinical Samples

Primers were designed based on PanEV EV-D68 sequences obtained in this study with an aim to generate two overlapping PCR products for whole-genome sequence determination by NGS. The nearly complete genome of EV-D68 strain from four respiratory samples; CLI-B3-55, 60, 77, and 78 were determined by NGS analysis of overlapping PCR products. Primers D68-WGF2_Mar2019 (5′-CCCACGTGGCGGCTAGTACTCTGG-3′) and D68_3757R_Mar2019 (5′-GTTCCATRGCATCRGTATCTTAACCA-3′) were used to generate a PCR product covering the 5′-end half of the genome and primers D68_uniIF_Mar2019 (5′-GGRGTAATWGGTCTTCTYACAGCAGG-3′) and D68_WGR2_Mar2019 (5′-GAAAGTAACTRYAACTTGGGTTTCAATTAGAG-3′) were used to generate a PCR product covering the 3′-end of the genome. Finally, the two overlapping contigs, containing reads mapping to the EV-D68 sequences were assembled to produce the whole-genome sequence as described before [19]. DNA amplification/purification conditions and NGS analysis to obtain consensus sequences were the same as those for entire-capsid RT-PCR amplification described in Section 2.3 and Section 2.5, respectively.

### 2.5. Preparation of Sequencing Libraries, Quality Trimming of NGS Reads, and Generation of EV Sequence Contigs

Processing and analysis of NGS data were performed using Geneious R10 software (Biomatters, Auckland, New Zealand) as described before [10]. Briefly, sequencing libraries were prepared using Nextera XT reagents and sequenced on a MiSeq using a 2 × 250 paired end v2 Flow Cell and manufacturer’s protocols (Illumina, CA, USA). Data were filtered using a custom workflow [10]. Sequence contigs were built by reference-guided assembly using a curated enterovirus sequence database and stringent assembly conditions: minimum 50 base overlap, minimum overlap identity of 98%, maximum 2% mismatches per read, allowing up to 15% gaps and both pair reads mapping. The same bioinformatics pipeline was used to generate both entire-capsid and whole-genome sequence contigs. Relevant FASTQ files used in this study are available from the NCBI Short Read Archive under project code PRJNA643298.

### 2.6. Viral Genome Analysis and Identification of EV Serotypes

EV sequences obtained in this study from both Sanger and NGS analysis, were compared to those available in the GenBank database using Geneious R10 software (Biomatters, Auckland, New Zealand) as described elsewhere [10,11]. The closest virus relatives to each of the EV final consensus sequences were identified using the RIVM and BLAST online sequence analysis tools and EV serotypes were assigned based on their VP1 sequence. Sequences generated for this paper are available from NCBI sequence database with GenBank numbers (MT641353-MT641450).

### 2.7. mECRA Followed by VP3-VP1 Nested PCR

For a subset of samples (*n* = 49) an alternative protocol based on Sanger sequencing was also tested. For this purpose, the mECRA PCR product was diluted 1:20 in nuclease free sterile water and 5 microliters of the diluted product was used as a template for nested PCR using pan-enterovirus published primers 222 and 224 that generate a 762 bp nucleotide PCR product from the VP3-VP1 coding region [8,20]. PCR reactions were prepared using Dream TaqTM hot start PCR master mix (Thermofisher Scientific, Waltham, MA, USA) with 0.4 µM of forward and reverse primers. Reaction conditions were 94 °C for 2 min plus 35 cycles of 94 °C for 15 s, 55 °C for 30 s, and 68 °C for 2 min with a final extension step of 68 °C for 5 min. Amplified products were purified using AMPure XP magnetic beads (Beckman Coulter, Brea, CA, USA) and sequenced by the Sanger method. The Sanger VP1 sequences were then compared with those obtained by NGS to assess sequence similarity.

### 2.8. Statistical Analysis

Statistical analyses were carried out using GraphPad Prism 8.1.2. Descriptive statistics of cycle threshold (*C*t) values gave minimum, maximum and percentile values. For comparison of independent Ct values between groups, the non-parametric Mann–Whitney U test was applied. The sensitivity and specificity of the mECRA-NGS method with respect to the Sanger conventional approach for each EV species were calculated as the probability that the NGS method produced a positive result when the result of the Sanger method was positive and the probability that the NGS result was negative when the Sanger result was negative, respectively. 

## 3. Results

### 3.1. EV Serotype Distribution in Clinical Samples and Comparison between Conventional and NGS Typing Methods

Full details of the comparison of EV typing results between the conventional Sanger method and the new NGS approach are shown in Appendix A and summarized in Table 1 and Table 2.

There was agreement in the serotypes identified in the 83 samples that were typed by both methods. The entire-capsid genomic NGS analysis of 112 clinical samples from England identified 94.74% (18/19), 93.75% (45/48), 100% (7/7), and 100% (13/13) of the EV-A, EV-B, EV-C, and EV-D strains that were identified by Sanger sequencing, respectively. These overall sensitivity values of the NGS method had associated 95% confidence intervals of 73.97% to 99.87%, 82.80% to 98.69%, 59.04% to 100% and 96.34% to 100% for Enterovirus A, B, C and D, respectively. In addition, the NGS method produced an EV typing result in 9 of previously untyped samples; 4 respiratory samples (human rhinovirus C), 4 CSF samples (2 EV A- CVA10, EV B-E6, E25), and 1 stool sample (EV B- E6). Furthermore, an additional EV serotype was found in 6 samples by NGS. In particular, 5 stool samples (20%) were found to contain an additional EV serotype to the one that was originally identified by the conventional method. In summary, 2 EV-A, 7 EV-B, and 2 EV-C strains were identified by the NGS method in samples that were not positive for these viruses by Sanger sequencing (giving specificities compared with Sanger sequencing of 97.85%, 89.06%, and 98.10% for these EV species, respectively). Sixteen samples untyped by the Sanger method remained untyped with the NGS method. EV serotype distribution showed predominance of EV-B species in blood (83.3%), CSF (84.6%), and stool (43.3%) samples, while respiratory samples showed a more complex distribution of EV species with EV-A (18.9%), EV-B (10.8%), EV-C (8.1%), EV-D (29.7%), and rhinovirus (10.8%) (Figure 1). Importantly, newly emerging EV serotypes such as EV-D68, EV-C104, EV-C105, EV-A89 were detected by both the Sanger and NGS methods.

### 3.2. Comparison of EV-qPCR Ct Values with Typing Results Using Sanger or NGS Methods

The median Ct values of EV RT-qPCR results for samples successfully typed by both methods was significantly lower than that for samples that were not typed (24.96 ± 4.96 vs 32.06 ± 5.10; *p* < 0.0001, Mann–Whitney U test) (Figure 2) indicating that inability to type EVs in samples is at least partly due to the low EV concentration levels present. 

However, although median Ct values of EV-qPCR results for samples that failed to type using the NGS approach were higher than those for samples that were untyped with the conventional method, these differences were not statistically significant (32.04 + 5.64 vs 26.95 + 5.27; *p* = 0.2141, Mann–Whitney U test). Interestingly, the three samples showing the lowest Ct values among samples untyped with the conventional method were identified as Rhinovirus C by the NGS method, showing that both the EV-qPCR and NGS methods can detect such viruses. 

### 3.3. mECRA Nested Approach for Generating Enterovirus Typing Information

The mECRA amplicon can be used as template to perform a nested PCR with either primers specific for selected EV serotypes as we have shown for multiple EV serotypes found in wastewater samples [11] or pan-enteroviruses primers targeting the VP3-VP1 genomic region [8,20]. This approach would be an alternative to the conventional Sanger sequence typing protocol described in Section 2.2, allowing a wider sequence window (approximately 300 vs 700 nucleotides) and would be more adequate for laboratories having limited access to NGS facilities or as an initial step to identify serotypes of interest for further sequencing. To test this approach with clinical samples, a random subset (*n* = 49) of mECRA amplicons was diluted 1:20 for amplifying VP3-VP1 gene segment using published PanEV primers 222 and 224 [8] followed by Sanger sequence analysis of the VP3-VP1 product. The results showed a 100% concordance for typing results determined by the mECRA-NGS and mECRA-nested Sanger methods. As expected, phylogenetic analysis showed 100% similarity between nucleotide sequences of all EV-A, -B, -C, and -D strains obtained by both methods (Figure 3) confirming the suitability of our NGS analysis pipeline.

### 3.4. Entire-Capsid Sequences Obtained for Uncommon Enterovirus

The study sample set included clinical samples with uncommon EV strains (defined as those from EV serotypes for which <15 whole genomes are reported in the NCBI data base). Entire-capsid genome sequences were obtained for 15 of such strains and their closest relatives in the NCBI sequence database were identified using BLAST (Table 3). Importantly, many of these uncommon EV strains (*n* = 9, 47.37%) were found in critical clinical samples from neurological syndromes like CSF.

### 3.5. Whole-Genome Determination of EV-D68 Strains from Clinical Samples

The nearly complete genome sequences of four EV-D68 strains (from nt 36 to nt 7326 based on numbering of EV-D68 Fermon reference sequence with GenBank number AY426531) were obtained from respiratory samples CLI-B3-55, 60, 77, and 78 as described in Material and Methods. The NGS analysis produced contigs with high sequence coverage throughout the genome (Appendix A) generating consensus sequences of 7253 nucleotides in length and very high sequence homogeneity as judged by single nucleotide polymorphism analysis. The four EV-D68 sequences were very similar between them and nearly identical (>99.8% sequence similarity) to previously sequenced EV-D68 clinical isolates obtained between 3 and 25 September 2018 in Spain and France (Accession Nos. MN245409, MN245412, MN245414, MN245425, MT789741, MT789744 and MT789748) during the same period as the clinical samples described from England. The whole-genome EV-D68 sequences are available from NCBI sequence database with GenBank numbers MW664343-MW664346

## 4. Discussion

This study describes the use of a rapid detection method, in that it does not require the use of cell culture methods, for the identification and characterization of EVs present in different types of clinical samples including CSF, respiratory, stool, skin, and blood. The main aim of the study was to evaluate the suitability of this molecular method based on NGS analysis of DNA amplicons obtained by entire-capsid region RT-PCR amplification (mECRA) directly from RNA extracted from clinical samples.

The mECRA-NGS method shown here produced highly concordant typing results with those from the conventional partial VP1 Sanger sequencing assay used across diagnostic laboratories. We were able to sequence multiple EV serotypes in a collection of 112 clinical samples from England. The overall sensitivity of this method to detect species A, B, C, and D EVs was 94.74%, 93.75%, 100%, and 100%, respectively, compared to the Sanger sequencing conventional method. The mECRA-NGS method has clear additional benefits due to the length of nucleotide sequences obtained and the ability to sequence EV mixtures. Using NGS, we could generate information of about 4000 nt. spanning the entire-capsid coding region. Benefits of compiling sequence data from a larger fragment are clear, such as monitoring mutations from existing circulating strains, potential association of genomic changes with specific clinical manifestations, updating sequences of detection and/or typing primers and structure modelling to understand viral antigenic properties but, besides all, establishing accurate spatial-temporal associations between clinical isolates which sometimes might be limited with the short 300 nt VP1 sequence provided by the Sanger method. An additional advantage of the mECRA-NGS approach relative to the Sanger conventional method, is its ability to identify EVs in mixtures, relatively common in stool samples (Table 1 and Table 2), giving more in-depth information of the actual causative agent/s of the disease. Overall, 2 EV-A, 7 EV-B, and 2 EV-C strains were identified by the NGS method in samples that were not positive for these viruses by Sanger sequencing. Another utility of the mECRA sequence data is that it can be used to expand the nucleotide sequencing analysis to generate whole-genome sequences of target EV serotypes as we have shown before [18] and we show here for few EV-D68 isolates.

The rapid and accurate identification of EVs in clinical samples will contribute to the clinical diagnosis of diseases associated with EV infections. There is evidence of EV serotypes causing severe disease, including neurological complications, such as EV-A71 and EV-D68 causing outbreaks associated with polio-like paralytic cases in recent years [4,6,22] to the extent that EV-A71 vaccines are now used in China where EV-A71-associated neurological disease is more prevalent [23]. Other NPEV serotypes could be the causative agents of acute flaccid paralysis (AFP) cases as they are frequently detected and isolated during laboratory surveillance for poliomyelitis. In Europe, NPEV surveillance is focused on hospital infections with more severe presentations. Hence, EV infections are often underdiagnosed and typing data are incomplete. In order to improve EV diagnostics, collate data on severe EV infections and monitor the circulation of EV types, a European non-polio enterovirus network (ENPEN) has recently been established and has published recommendations for enterovirus diagnostics [7]. Using our method, we were able to sequence several strains from serotypes EV-A71, EV-D68, CV-A6, EV-C104, EV-C105, and E-30, recently associated with severe disease in humans [4,6,24,25,26]. 

In a resource limited setting, where NGS access is more restrictive, the mECRA product can be used as a template for nested PCR reactions using EV serotype/genotype specific primers as we have shown before [11] or degenerate primers targeting the VP3-VP1 genomic region as we show here. Our methods were recently assessed in a multi-center study that evaluated the sensitivity and specificity of currently used commercial and in-house diagnostic and typing assays using in vitro RNA transcripts representing the four EV species (EV-A71, echovirus 30, coxsackie A virus 21, and EV-D68). Both the mECRA-NGS and mECRA-nested methods showed high sensitivity and specificity and, as well as other in-house assays, showed significantly greater detection frequencies of the low copy RNA preparations than commercial assays [27]. This is an important finding because viral loads are relatively low in critical samples such as CSF in patients presenting with meningitis or encephalitis [28,29]. 

Admittedly, any amplification-based method such as the mECRA assay described here may contribute to the bias toward specific strains or serotypes and may risk missing some strains due to mismatches in primer-binding sequence regions. However, we have extensively used the mECRA-NGS approach to sequence EVs present in sewage samples collected in the UK, Pakistan, Senegal, and Nigeria identifying multiple EV serotypes showing the complex EV circulation patterns in humans [11,16,18,19]. So far, we have been able to sequence EV strains from 102 of the 110 different EV serotypes that have been described to date. 

## 5. Conclusions

We have shown that using NGS analysis of amplicons produced by mECRA allows the sensitive and specific detection of multiple EV serotypes present in clinical samples. The method has various advantages over the Sanger sequencing conventional method based on the analysis of short VP1 sequences (300 nt). The mECRA-NGS approach provides a large amount of sequencing information, that of 4000 nucleotides spanning the entire-capsid coding region and can identify several EV strains in mixtures. It is expected that NGS methods, such as the one described here, will replace current diagnostic methods in the near future, which will result in the better understanding of EV transmission patterns in different human populations.

## Figures and Tables

**Figure 1 viruses-13-00641-f001:**
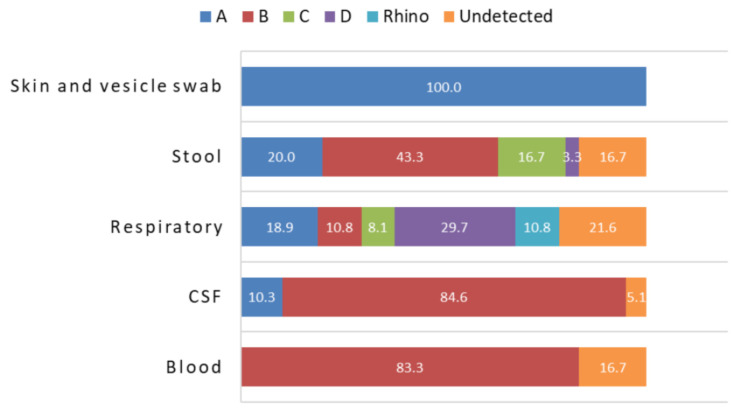
Distribution of EV species by sample type in 112 clinical samples from England determined by NGS analysis.

**Figure 2 viruses-13-00641-f002:**
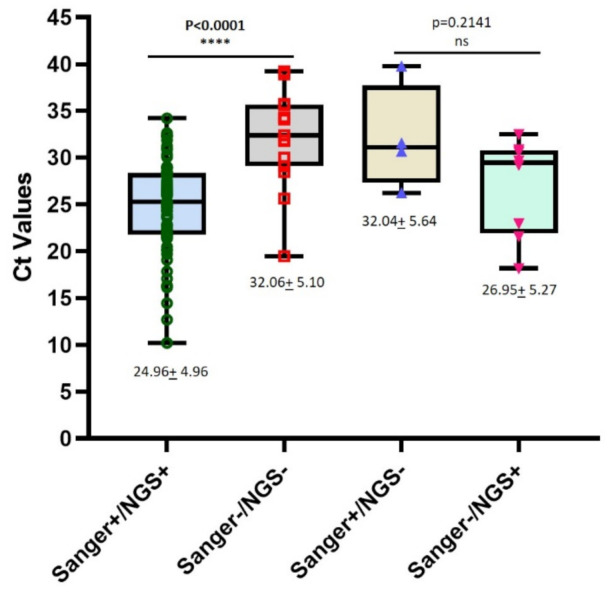
Box charts representing comparisons of EV RT-qPCR Ct values between samples with positive/negative EV typing results using conventional PCR followed by Sanger sequencing and mECRA RT-PCR followed by NGS analysis. Median and lower/upper quartiles are shown as lines inside and outside top/bottom edges of the box, respectively. Mann–Whitney U test was used to determine statistical significance; **** showing statistically significant result and ns showing statistically non-significant result.

**Figure 3 viruses-13-00641-f003:**
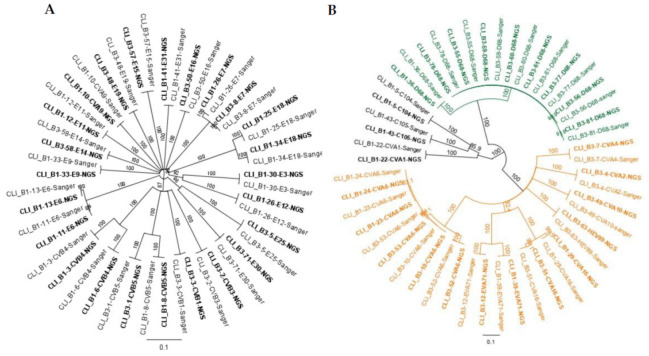
Similarity between nucleotide sequences from EVs present in clinical samples using NGS analysis (bold text) or nested VP3-VP1 Sanger sequencing (plain text). (**A**) species B EVs. (**B**) species A (orange), C (black), and D (green) EVs. Phylogenetic tress were generated using the Maximum Likelihood method and Tamura-Nei model using MEGA X software. The percentage of replicate trees in which the associated taxa clustered together in the bootstrap test (1000 replicates) are shown next to the branches.

**Table 1 viruses-13-00641-t001:** Identification of enteroviruses (EVs) in 112 clinical samples from England using next generation sequencing (NGS) analysis compared with conventional Sanger sequencing by type of clinical sample.

Type of Sample	No. of Samples (*n* = 112)	Enterovirus Sanger (+) NGS (+)	Enterovirus Sanger (−) NGS (−)	Enterovirus Sanger (−) NGS (+)	Enterovirus Sanger (+) NGS (−)	>1 Enterovirus Detected by NGS
CSF	39 (34.82%)	31 (79.48%)	2 (5.12%)	4 (10.25%)	2 (5.12%)	0
Blood	6 (5.36%)	4 (66.66%)	1 (16.67%)	0	1 (16.67%)	0
Respiratory	36 (32.14%)	24 (66.67%)	8 (22.22%)	4 (11.11%) ^1^	0	1 (2.78%)
Stool	25 (22.32%)	18 (72%)	5 (20%)	1 (4%)	1 (4%)	5 (20%)
Skin/Vesicle Swab	4 (3.57%)	4 (100%)	0	0	0	0
Not Specified	2 (1.79%)	2 (100%)	0	0	0	0
Total	112	83	16 (20) ^2^	9 (5) ^2^	4	6

^1^ Rhinovirus C was identified in four samples. ^2^ Excluding Rhinovirus C in the calculation.

**Table 2 viruses-13-00641-t002:** Identification of EVs in 112 clinical samples from England using NGS analysis compared with conventional Sanger sequencing by EV species ^1^.

Sanger Result	NGS Result (No. of Samples)								
	Ent A	Ent B	Ent C	Ent D	Ent A + B	Ent B + C	Ent B + D	Rhino C	Negative
Ent A	**16**				2				1
Ent B		**43**				2			3
Ent C			**7**						
Ent D				**11**			2		
Negative	2	3						4	**16**

^1^ Concordant result by NGS and Sanger sequencing are highlighted in bold.

**Table 3 viruses-13-00641-t003:** Entire capsid sequences and genome coverage data of uncommon EVs from this study.

Clinical Sample	Accession No.	Closest Relative from NCBI Sequence Database					
		Accession No.	% Identity	Year	Country	Serotype	Whole Genomes Available
CLI-B3-63/Stool	MT641439	AY697459	89.61	2000	Bangladesh	Enterovirus A89	2
CLI-B2-13/CSF	MT641365	LC120911	87.43	2015	China	Echovirus E2	3
CLI-B1-14/CSF	MT641366	HM775882	82.77	2006	South Korea	Echovirus E5	3
CLI-B1-28/Stool	MT641379	MH144602	89.98	2011	India	Echovirus E12	7
CLI-B3-9/CSF	MT641402	MK086261	95.03	2015	France	Echovirus E13	11
CLI-B3-43/CSF	MT641423	FJ868345	86.58	2004	Australia	Echovirus E14	7
CLI-B3-58/CSF	MT641435	MF990302	84.02	2016	Ethiopia	Echovirus E14	7
CLI-B3-57/CSF	MT641434	KU133611	86.9	2012	Russia	Echovirus E15	1
CLI-B3-50/CSF	MT641428	KP289436	93.41	2013	China	Echovirus E16	3
CLI-B1-42/Stool	MT641387	MH933855	79.5	2014	Cameroon	Echovirus E20	15
CLI-B1-44/Stool	MT641389	MH933854	80.08	2014	Cameroon	Echovirus E20	15
CLI-B1-20/CSF	MT641371	LC120936	91.86	2015	China	Echovirus E21	1
CLI-B1-41/CSF	MT641386	JN203962	82.46		India	Echovirus E31	1
CLI-B1-17/Stool	MT641370	MG571859	87.7	2015	Venezuela	Coxsackievirus A1	6
CLI-B1-22/Stool	MT641373	MH361027	87.39	2015	UK	Coxsackievirus A1	6
CLI-B3-17/Stool	MT641408	MF990306	82.36	2016	Ethiopia	Coxsackievirus A17	8
CLI-B1-5/Resp	MT641356	MN481403	98.01	2018	Belgium	Enterovirus C104	12
CLI-B1-43/Uns	MT641388	KX276189	97.11	2014	USA	Enterovirus C105	10
CLI-B1-46/Resp	MT641392	KM880100	97.9	2011	Italy	Enterovirus C105	10

## Data Availability

Nucleotide sequences determined in this study are available from NCBI sequence database with GenBank numbers (MT641353-MT641450, MAW664343-MW664346). Relevant FASTQ files used in this study are available from the NCBI Short Read Archive under project code PRJNA643298.

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
