# Peer review of "Detection and Typing of Human Enteroviruses from Clinical Samples by Entire-Capsid Next Generation Sequencing"

_viruses, 2021, doi:10.3390/v13040641_

Round 1

Reviewer 1 Report

The authors examined a direct detection method for EVs based on sanger sequencing and next generation sequencing (NGS) analysis of the entire capsid coding region. 112 clinical samples were analyzed. There was high concordance between the results obtained by the new NGS approach and those from the Sanger method. The NGS method identified 5 EVs in previously untyped samples as well as additional viruses in some samples, indicating co-infection. Information from VP1 capsid and whole-genome sequences is critical to help identifying the genetic basis for changes in viral properties and establishing accurate spatial-temporal associations between EV strains of public health relevance.

Overall the referee has no critical comments regarding of performance of the investigations. The experimental procedures are OK and the results are adequate for publication.

Author Response

We thank the reviewer for his/her positive comments on our paper.

Reviewer 2 Report

The authors used the novel NGS analysis of amplicons produced by mECRA and Sanger sequencing to compare the type, sensitivity and specificity of enteroviruses (EVs) in different samples. The authors revealed that 83 out of 112 samples analyzed by NGS analysis of amplicons produced by mECRA and Sanger sequencing were consistent, and NGS analysis of amplicons produced by mECRA additionally analyzed the virus types that could not be analyzed by Sanger sequencing. NGS analysis of amplicons produced by mECRA also analyzed 4 respiratory samples (human rhinovirus C), 4 cerebrospinal fluid samples (2 EV A-CVA10, EV B-E6, E25) and 1 fecal sample (EV B-E6) that could not be analyzed by Sanger sequencing. Amplicons produced by mECRA can be a novel method for diagnosing enteroviruses and identifying enterovirus types in the environment.  After reading the whole article, my first impression is that this is an unfinished first draft of the article. There are many typos and incomplete sentences in the whole article

Line 42-43. Please provide a reference for this description.

Line 45. Please re-write this sentence since there is both present and past tense in the sentence.

Table 1. There is EV Sanger (+) NGS (-). Why this is written as EV instead of Enteroviruses ?

Line 76: the reference is needed.

Line 239. I don’t understand why the authors want to compare the Ct value obtained from qPCR to the result using Sanger or NGS. What’s the meaning of comparison of these two methods?

Author Response

We thank reviewer 2 for his/her suggestions that we have addressed below. Changes can be reviewed using the tracked version of the modified manuscript.

Comment from reviewer

The authors used the novel NGS analysis of amplicons produced by mECRA and Sanger sequencing to compare the type, sensitivity and specificity of enteroviruses (EVs) in different samples. The authors revealed that 83 out of 112 samples analyzed by NGS analysis of amplicons produced by mECRA and Sanger sequencing were consistent, and NGS analysis of amplicons produced by mECRA additionally analyzed the virus types that could not be analyzed by Sanger sequencing. NGS analysis of amplicons produced by mECRA also analyzed 4 respiratory samples (human rhinovirus C), 4 cerebrospinal fluid samples (2 EV A-CVA10, EV B-E6, E25) and 1 fecal sample (EV B-E6) that could not be analyzed by Sanger sequencing. Amplicons produced by mECRA can be a novel method for diagnosing enteroviruses and identifying enterovirus types in the environment.  After reading the whole article, my first impression is that this is an unfinished first draft of the article. There are many typos and incomplete sentences in the whole article

Response by authors

We are sorry the reviewer feels there are many errors in the manuscript and apologize for this. We have gone through the manuscript and corrected few errors throughout.

Comment from reviewer

Line 42-43. Please provide a reference for this description.

Response by authors

We have added a reference for this description.

Comment from reviewer

Line 45. Please re-write this sentence since there is both present and past tense in the sentence.

Response by authors

We think this sentence is correct and have left it as it was.

Comment from reviewer

Table 1. There is EV Sanger (+) NGS (-). Why this is written as EV instead of Enteroviruses?

Response by authors

Enterovirus instead of EV has been written in all columns of Table 1 to maintain consistency.

Comment from reviewer

Line 76: the reference is needed.

Response by authors

There is no need for a reference as the full method has been described which is specific for this paper.

Comment from reviewer

Line 239. I don’t understand why the authors want to compare the Ct value obtained from qPCR to the result using Sanger or NGS. What’s the meaning of comparison of these two methods?

Response by authors

The idea is to be able to compare the sensitivity of both methods based on estimated virus concentrations. We have used  the Ct values as an estimate of virus concentration with higher values corresponding to lower virus titres as used in many other studies.

Reviewer 3 Report

Reviewer’s report

Manuscript:

Direct Detection and Typing of Human Enteroviruses from Clinical 2 Samples by Entire-Capsid Next Generation Sequencing

The manuscript compare NGS method with conventional methods (Sanger sequencing in the 5’UTR of VP1 gene and 762 bp in the VP3-VP1 genes) used to detect Non-Polio Enteroviruses in clinical samples.

  1. Tilte: line 2 : Authors mention direct detection: however, they needed to go through RNA extraction followed by RT-PCR to amplify the 4000nt than preparation of libraries for NGs

I think that the term “direct” is not appropriate   

  1. line 18: the 112 is directly after the point, it is more appropriate to spell it with letters or to add “A total” this remark is applicable for all the text.
  2. Line 25: authors mention 100% of sensitivity with a too large interval of confidence (04% to 100%) please verify the interval.
  3. line 33-34: Authors mention as keywords: wastewater however, there are no wastewater samples tested in the present study and direct detection which is not appropriate
  4. line 45: Picornaviridae should be in italic
  5. line 76 and 94: please add references
  6. lines 102-104: please uniform the writing of the primer sequences. It is applicable for all the primers
  7. line 201 page 5: table 1: the total of percentage of detected EV in CSF in line 1 are not 100% please verify

Please adjust the table layout

  1. line 202: the perfect agreement is for 77 positives and 16 negatives and not 83 as author mentioned, please verify
  2. line 211: The numbers of EV-A and EV-B in the text and in table 2 seem not the same, please verify, same goes for line 364-365.
  3. Line 213: Please specify how do you obtained the specificity and sensitivity rates of the method (please specify the method of calculation).
  4. Line 216-218: The percentages in the text and in Fig 1 are not the same, please verify.
  5. Lines 242: there are “minus” missing in the intervals same for line 271
  6. Paragraph 3.3: the main aim of the present study is to compare NGS method with conventional methods please specify the added value in this work of using mECRA nested approach
  7. Figure 3: it is obvious to obtain 100% of similarity of sequences once it is the same patients and samples compared
  8. Figure 4: what is the added value of Fig 4?
  9. Line 328: Please explain how authors evaluated the fact that the method is rapid and direct.
  10. Line 333: Authors did not specify the sensitivity of the method in result section.
  11. Discussion: Authors mention that NGS method is more efficient in detecting EV-D68 and EV-A71. However, according to the table S1 the EV-D68 were detected by both methods for all samples and EV-A71 was detected for one sample by Sanger method and not by NGS (lines 336-350).
  12. Please review the discussion to reflect more the aim and the result of the study.

Author Response

We thank reviewer 3 for his/her thorough review that will help us improve our paper. Changes can be reviewed using the tracked version of the modified manuscript.

Comment from reviewer

  1. Tilte: line 2 : Authors mention direct detection: however, they needed to go through RNA extraction followed by RT-PCR to amplify the 4000nt than preparation of libraries for NGs

I think that the term “direct” is not appropriate   

Response by authors

We don’t think this is a major concern and a purely semantic issue. We mention “direct” to reflect the absence of major processes such as cell culture amplification to obtain sequencing data from samples although we admit that there might be “more direct” detection methods sequencing RNA directly (which also require their own processes). Indeed, previous publications define similar methods as “direct detection”, e.g. Arita et al. Development of an efficient entire-capsid-coding-region amplification method for direct detection of poliovirus from stool extracts. J Clin Microbiol 2015, 53, (1), 73-8. However, we are happy to remove the word “direct” and have modified the title and text accordingly.

Comment from reviewer

  1. line 18: the 112 is directly after the point, it is more appropriate to spell it with letters or to add “A total” this remark is applicable for all the text.

Response by authors

We have modified the manuscript following your suggestion.

Comment from reviewer

  1. Line 25: authors mention 100% of sensitivity with a too large interval of confidence (04%to 100%) please verify the interval.

Response by authors

We could not see this error in either the original word or pdf version of the manuscript. The text in the manuscript states “(59.04% to 100%)” not 04% to 100% as the reviewer mentions. Still, it is true that this interval for enterovirus species C is larger than that for other species. This is due to the low number of enterovirus C isolates present in the sample set.

Comment from reviewer

  1. line 33-34: Authors mention as keywords: wastewater however, there are no wastewater samples tested in the present study and direct detection which is not appropriate

Response by authors

We have modified the manuscript following your suggestion.

Comment from reviewer

  1. line 45: Picornaviridae should be in italic.

Response by authors

We have modified the text following your suggestion.

Comment from reviewer

  1. line 76 and 94: please add references

Response by authors

There is no need for a reference as the full method has been described which is specific for this paper.

Comment from reviewer

  1. lines 102-104: please uniform the writing of the primer sequences. It is applicable for all the primers

Response by authors

We have modified the text following your suggestion.

Comment from reviewer

  1. line 201 page 5: table 1: the total of percentage of detected EV in CSF in line 1 are not 100% please verify

Please adjust the table layout

Response by authors

This is incorrect. The total of percentage by adding figures from columns 3 to 6 for each row should be 100% (or very close) which it is the case with the Table 1 as it is.

Comment from reviewer

  1. line 202: the perfect agreement is for 77 positives and 16 negatives and not 83 as author mentioned, please verify

Response by authors

There was perfect agreement in the serotypes identified in the 83 samples that were typed by both methods. This means that the typing result obtained by the NGS method for those 83 samples matched the result obtained by the Sanger method. However, additional viruses were found in some of those 83 samples using the NGS method which is not possible with the Sanger method as it can only produce a single sequence (either a clear sequence or a mixed unreadable sequence). We have decided to remove the word perfect to avoid confusion.

Comment from reviewer

  1. line 211: The numbers of EV-A and EV-B in the text and in table 2 seem not the same, please verify, same goes for line 364-365.

Response by authors

The reviewer is correct. Thank you for spotting this typing error. We have reviewed the raw data and found out that data on Table 2 are correct. Based on Table 2’s results, the correct text should say:

“2 EV-A, 7 EV-B and 2 EV-C strains were identified by the NGS method in samples that were not positive for these viruses by Sanger sequencing”

The number for EV-A was a typo in the manuscript so the value for specificity remains the same. The number for EV-B was an error, we used 5 instead of the real number 7,  and so the actual value for specificity changes slightly. This does not change any of the main outcomes and conclusions of the paper. The manuscript has been corrected accordingly in the Abstract and 3.1 Results section.

Comment from reviewer

  1. Line 213: Please specify how do you obtained the specificity and sensitivity rates of the method (please specify the method of calculation).

Response by authors

We have used standard definitions for sensitivity and specificity used in studies comparing two different methods. The sensitivity and specificity of the mECRA-NGS method with respect to the Sanger conventional approach for each enterovirus species were calculated as the probability that the NGS method produced a positive result when the result of the Sanger method was positive and the probability that the NGS result was negative when the Sanger result was negative, respectively. We have added this definition to section 2.8 in the Materials and Methods section.

Comment from reviewer

  1. Line 216-218: The percentages in the text and in Fig 1 are not the same, please verify.

Response by authors

The reviewer is correct. These were typographical errors and have been corrected in the modified manuscript.

Comment from reviewer

  1. Lines 242: there are “minus” missing in the intervals same for line 271

Response by authors

We have modified the text following your suggestion.

Comment from reviewer

  1. Paragraph 3.3: the main aim of the present study is to compare NGS method with conventional methods please specify the added value in this work of using mECRA nested approach

Response by authors

As we state in the manuscript, the mECRA-nested approach would be an alternative in laboratories having limited access to NGS facilities or as an initial step to identify serotypes of interest for further sequencing. A common criticism of NGS methods is their cost and technical difficulties to be applied in some countries/regions with low resources. In those cases, both the conventional and the mECRA nested approaches would at least provide critical information on EVs circulating in the population with the latter mECRA-nested approach offering an expanded sequencing window for a more in-depth spatial-temporal analysis of EV isolates. This explanation is already covered in the manuscript (section 3.3 and fourth paragraph of the Discussion section).

Comment from reviewer

  1. Figure 3: it is obvious to obtain 100% of similarity of sequences once it is the same patients and samples compared

Response by authors

We think it is “expected” rather than “obvious” that there is similarity between viral sequences from the same samples using different sequencing methods. However, NGS analysis use different molecular processes to Sanger sequencing and involve analysis of nucleotide sequences from thousands of small DNA fragments which require the use of bioinformatics pipelines to recover accurate consensus sequences. Although NGS methods are already well established for sequencing viral genomes, there is still merit in comparing NGS sequences with those from Sanger sequencing analysis to confirm the suitability of the NGS approach and the fact that different sequencing methods are possible to suit different situations. We have modified the text in section 3.3 slightly to reflect this explanation.

Comment from reviewer

  1. Figure 4: what is the added value of Fig 4?

Response by authors

Readers/reviewers normally request information on the quality of NGS analysis used to determine consensus sequences which includes sequencing reads coverage across the genome. We agree with the reviewer that Figure 4 is not essential and have moved it to the supplementary section.

Comment from reviewer

  1. Line 328: Please explain how authors evaluated the fact that the method is rapid and direct.

Response by authors

We have already discussed this above. We mention “direct” to reflect the absence of major processes such as cell culture amplification to obtain sequencing data from samples and in this sense, the method is also rapid, allowing the obtention of nucleotide sequences form large DNA amplicons (entire-capsid or whole-genome) in a single process. We agree that there might be “more direct” detection methods sequencing RNA directly (which also require their own processes). We are happy to remove the word “direct” and have modified the title and text accordingly.

Comment from reviewer

  1. Line 333: Authors did not specify the sensitivity of the method in result section.

Response by authors

The sensitivity of the mECRA-NGS method with respect to the Sanger conventional approach, calculated as the probability that the NGS method produced a positive result when the result of the Sanger method was positive, was specified in section 3.1 of the Results section. We have clarified this and added 95% Confidence Intervals for these values in the 3.1 paragraph.

Comment from reviewer

  1. Discussion: Authors mention that NGS method is more efficient in detecting EV-D68 and EV-A71. However, according to the table S1 the EV-D68 were detected by both methods for all samples and EV-A71 was detected for one sample by Sanger method and not by NGS (lines 336-350).

Response by authors

We do not state in the manuscript that “the NGS method is more efficient in detecting EV-D68 and EV-A71”. Our main conclusion as written in the Discussion’s second paragraph is that “The mECRA-NGS method shown here produced highly concordant typing results with those from the conventional partial VP1 Sanger sequencing assay used across diagnostic laboratories”. The mECRA-NGS method has clear additional benefits due to the length of nucleotide sequences obtained and the ability to sequence EV mixtures as we discuss in detail in the same paragraph. However, we have not shown that the NGS method is statistically better or worse in detecting EV-D68 and EV-A71 in terms of sensitivity than the Sanger conventional method. We have modified the text slightly in the third paragraph of the Discussion to avoid misunderstanding of our conclusions.

Comment from reviewer

  1. Please review the discussion to reflect more the aim and the result of the study.

Response by authors

We normally try to avoid repeating too much text describing aims in the study that are detailed in the Introduction section in the Discussion section but we have added a short sentence in the first paragraph of the Discussion to describe the main aim of our study. We have also reorganized the Discussion section swapping paragraphs 2 and 3 in the original version and re-writing some parts so it the whole section reads better.

Round 2

Reviewer 2 Report

I don't have any comment based on the new revised manuscript